# Multidrug-Resistant *Escherichia coli* from Raw Cow Milk in Namwala District, Zambia: Public Health Implications

**DOI:** 10.3390/antibiotics12091421

**Published:** 2023-09-08

**Authors:** Wizaso Mwasinga, Misheck Shawa, Patrick Katemangwe, Herman Chambaro, Prudence Mpundu, Ethel M’kandawire, Chisoni Mumba, Musso Munyeme

**Affiliations:** 1Department of Disease Control, School of Veterinary Medicine, University of Zambia, Lusaka P.O. Box 32379, Zambia; jejifpa@yahoo.com (P.K.); ethel.mkandawire@unza.zm (E.M.); cmumba@unza.zm (C.M.); mussomunyeme@gmail.com (M.M.); 2Hokudai Center for Zoonosis Control in Zambia, Hokkaido University, Lusaka P.O. Box 32379, Zambia; misheckshawa@ymail.com; 3Central Veterinary Research Institute, Ministry of Fisheries and Livestock, Lusaka P.O. Box 33980, Zambia; hermcham@gmail.com; 4Department of Environmental and Occupational Health, Levy Mwanawasa Medical University, Lusaka P.O. Box 33991, Zambia; prudencezimba@gmail.com

**Keywords:** AMR, *E. coli*, raw milk, Zambia

## Abstract

*Escherichia coli* (*E. coli*), a major foodborne disease-causing pathogen found in raw cow milk, has even far more reaching public health ramifications as it encodes for antimicrobial resistance (AMR). This study aimed to identify multidrug-resistant (MDR) *E. coli* from raw cow’s milk and evaluate their antimicrobial-resistant profiles. In total, 418 pooled raw cow milk samples were collected from milk collection centers and analysed using standard culture methods to isolate *E. coli.* Antimicrobial Susceptibility Testing (AST) was conducted using the Kirby Bauer disk diffusion method and PCR was used to identify cefotaxime (CTX) resistant genes. Overall isolation of *E. coli* was 51.2% (214/418) with MDR observed in 21% (45/214) of isolates across different antibiotic combinations. Resistance was observed towards ampicillin (107/214, 50%), tetracycline (86/214, 40.1%), trimethoprim/sulfamethoxazole (61/214, 28.5%), and amoxicillin/clavulanic acid (CTX) (50/214, 23.4%). Notably, 15% (32/214) resistance to CTX was observed, while 12.6% (27/214) exhibited resistance to imipenem. The *bla*_CTX-M_ and *bla*_TEM_ genes were detected in CTX-resistant isolates. The findings of MDR *E. coli* that harbour *bla*_CTX-M_ and *bla*_TEM_ genes in raw cow’s milk indicate serious public health risks for consumers.

## 1. Introduction

Increases in human population has been the major driver of agricultural transformation, leading to large-scale intensive production of farm animals such as poultry, pork, fish, and beef to meet the ever-increasing demand for dietary protein [1]. In Zambia, raw milk is considered an affordable and readily available source of dietary protein that is produced and consumed in large quantities in both rural and urban farming communities [2]. Although Zambia has both intensive and traditional milk production systems as part of the dairy value chain, the traditional farmers’ milk production sector is by far the most widespread and contributes significantly to national dairy production. However, the Zambian traditional milk production sector is characterized by poor food handling and sanitation practices. The sector also has inadequate primary-level animal handling shelters and equipment [3]. Apart from farm-level problems, at a national level, there are weak regulatory systems and lack of education for food handlers, and little or no sanitary and hygienic standards observed [3]. However, while extensive milk production is vital for alleviating poverty, the associated contamination with bacteria may result in debilitating foodborne diseases, which is further complicated by the consumption of pathogens present in the raw milk that encode antimicrobial resistant determinants [4]. 

Of major concern to human health is the issue of antimicrobial resistance (AMR) due to the use of antibiotics in livestock production. Antimicrobial use in livestock production and treatment has significantly increased the threat of AMR in humans [5,6,7]. AMR threatens the effective treatment of foodborne illnesses and has led to increased health costs, longer duration of illness, and increased mortality in humans and animals [8,9]. Raw milk is no exception and has been known to be fecally contaminated with many *Enterobacteriaceae*, especially *E. coli* [10,11]. Further, these *E. coli* species have been observed to develop resistance to antibiotics used for treating human and livestock diseases [12]. Furthermore, there has been an increase in the occurrence of extended-spectrum β-lactamase (ESBL)-producing *Enterobacteriaceae* in foods of animal origin, with CTX-M types being the most common [13,14].

The close interaction between humans and livestock presents a unique opportunity for AMR transmission across the ecosystem [15]. For instance, humans may acquire drug-resistant strains of animal origin by direct contact with food contaminated by animal wastes or indirectly from consuming products from these animals [4,16]. Further, the presence of AMR *E. coli* in the environment, such as water sources and grazing areas, can act as a source of infection and transmission for dairy cattle [17]. Furthermore, movement of animals for trade or slaughter between districts locally or cross-border trade plays a significant role in the spread of antimicrobial resistant *E. coli* [18]. Of critical importance to public health is resistance that is passed on between species due to horizontal gene transfer of antimicrobial-resistant genes (ARG) [19,20]. Such spread or transfer of ARG between livestock products and human microbial populations increases the risk of multidrug resistance amongst commensals as well as pathogens and may result in reduced efficacy of antibiotic treatment and lead to increased drug resistance. *E. coli* possesses genes that encode resistance to antibiotics. Henceforth, *E. coli* is a pathogen of key AMR concern. 

Despite research in food animals improving in contemporary times, studies on AMR microbes in milk are relatively rare in Zambia, especially in rural areas. For instance, in Zambia, reports of drug-resistant *Enterobacteriaceae* have generally focused on humans and poultry in urban areas [21,22,23]. However, the AMR situation in raw milk remains limited, particularly in rural areas where research is generally neglected. Namwala District in the southern part of Zambia is a typical paradigm of milk production in the country and thus creates a unique context for studying AMR in milk. This study investigates the phenotypic and genotypic AMR patterns among *E. coli* isolated from raw milk.

## 2. Results

### 2.1. Descriptors of E. coli Prevalence

Overall *E. coli* contamination from all four tested milk collection centers (MCCs) was 214/418, thereby giving a prevalence of 51%. MCC 4 had the highest proportion (70%) of *E. coli* contamination, while raw milk from MCC 2 had the least *E. coli* contamination (36%) (Table 1). 

### 2.2. Resistance Was Highest to Amoxicillin, Sulfonamides, and Tetracycline

The AMR patterns of all of the 214 tested *E. coli* isolates are shown in Figure 1. The highest resistance was observed towards ampicillin (107/214, 50%), followed by tetracycline (86/214, 40.1%), trimethoprim/sulfamethoxazole (61/214, 28.5%), and amoxicillin/clavulanic acid (50/214, 23.4%). Notably, 15% (32/214) of the isolates were resistant to CTX, while 12.6% (27/214) exhibited resistance to imipenem. In addition, there was also resistance to ciprofloxacin (13/214, 6.1%).

### 2.3. Prevalence of Multi-Drug Resistance (MDR)

From a total of 214 *E. coli* isolates, 21% (45/214) exhibited resistance to more than 3 drugs (MDR). Furthermore, two isolates (2/214) representing 0.9% were possibly extensively drug-resistant (XDR). Of the 45 MDR strains, 17.8% (8/45) displayed resistance to a drug combination involving AMC, AMP, CHL, TCY, and SXT.

### 2.4. bla_CTX-M_ and bla_TEM_ ESBL Genes Were Detected in the Isolates

Based on the disk diffusion method 32/214 (15%) *E. coli* isolates were CTX-resistant. The 32 isolates were further subjected to the broth microdilution test. Fourteen of the 32 CTX-resistant *E. coli* isolates showed reduced susceptibility to cefotaxime (MIC ≥ 2 μg/mL). The *bla*_CTX-M_ and *bla*_TEM_ genes were detected by PCR in five and six out of 14 beta-lactam resistant isolates, respectively (Table 2).

## 3. Discussion

In this study, multidrug-resistant *E. coli* were identified in raw milk sampled from four MCCs in Namwala district, Zambia. The prevalence of MDR *E. coli* in this study was determined to be 21% (45/214). This prevalence was lower compared to the 34.7% observed by Dowidar and Khalifa [24] but higher compared to 14.8% reported by Ngaywa et al. [25] in similar studies conducted in Egypt and Northern Kenya respectively [24,25]. Furthermore, the overall prevalence of *E. coli* in this study was higher, reaching 51.2% (214/418), when compared to 40% observed by Dowidar and Khalifa [24] in Egypt and the 33.3% reported by Knight et al. [10] in Western Zambia, respectively [10,24]. These findings may be attributed to factors such as antibiotic overuse, poor hygiene practices, and faecal contamination in the primary production stage, as observed in similar studies where MDR prevalence ranged from 14.8% to 34.7% and *E. coli* prevalence ranged from 33.3% to 40% [10,24,25]. Therefore, it is essential to address these underlying factors through antibiotic stewardship programs, enhanced hygiene practices and strict monitoring of primary production processes. Such measures can help mitigate the emergence and spread of antibiotic resistance, reduce contamination rates and safeguard both human and animal health.

The AMR profiles of the *E. coli* isolates in this study revealed resistance to commonly used antibiotics, including ampicillin AMP (50%), tetracycline TCY (40.1%), trimethoprim/sulfamethoxazole SXT (28.5%), amoxicillin/clavulanic acid AMC (23.4%) and cefotaxime CTX (15%) with 17.8% (8/45) multi-drug resistance (MDR) towards AMC, AMP, CHL, TCY, and SXT classes of antibiotics. Comparing these findings to the MDR pattern observed by Ngaywa et al. [25], similarities can be observed. Although the specific classes of antibiotics involved differ slightly, both studies identified the presence of MDR *E. coli* strains. In the study by Ngaywa et al. [25] the MDR pattern was observed towards TCY, AMP, AMC, and CTX classes of antibiotics. These similarities in MDR patterns between the two studies suggest the persistence of multidrug–resistant *E. coli* strains across different regions or settings. This is of public health importance since exposure to AMR *E. coli* provides an opportunity for the likelihood of spread of resistance to pathogenic *E. coli* thereby resulting in complicated treatment of infections [26]. Moreover it is worth noting that the antibiotics under investigation in this study namely AMC, AMP, CHL, TCY and SXT, are predominantly utilized as first-line treatments for both human and livestock infections [27,28]. Previous studies investigating AMR patterns of *E. coli* in raw milk have reported resistance to TCY, AMP, AMC, and CTX, with observations of MDR in 14.8% to 34.7% of the isolated *E. coli* strains [25,29]. The emergence of antibiotic resistance in these studies has been attributed to the misuse and overuse of antibiotics, particularly as these antibiotics are affordable and accessible, commonly employed to treat environmental mastitis and lumpy skin disease; both prevalent disease conditions within the dairy sector [30]. Furthermore, the affordability and accessibility of these antibiotics can be attributed to the significant influx of penicillin, tetracycline, and sulfonamide antibiotic classes into the country for their use as veterinary medicinal products [31]. 

A matter of great concern in this study was the presence of imipenem resistance in 12.6% of the *E. coli* isolates. Carbapenems including imipenem are currently considered as the last–resort drug class for the treatment of severe infections, especially when other antibiotics are ineffective. Although clinical carbapenem resistance is rare in Zambia [32,33], our study’s finding of imipenem resistance deserves attention and requires more investigation. Furthermore, it is noteworthy that some of the identified carbapenem- resistant isolates were also among the MDR strains that exhibited resistance to AMC, AMP, CHL, TCY, and SXT. 

Another significant finding in this study was the 15%(32/214) resistance of *E. coli* isolates to CTX on the phenotypic test and the presence of five *bla*_CTX-M_ and six *bla*_TEM_ genes on the genotypic test. This resistance to CTX (15%) was lower compared to the 44.4% observed by Demirci et al. [34] who conducted a similar study in Turkey and identified five *bla*_CTX-M_ and four *bla*_TEM_ genes out of ten ESBL positive strains [34]. ESBL enzymes are encoded mainly by *bla*_CTX-M_ and *bla*_TEM_ genes usually located on bacterial plasmids, contributing to resistance against broad-spectrum cephalosporins such as CTX [35]. The disparity observed between the phenotypic and genotypic tests could potentially be attributed to *E. coli* isolates harboring other ESBLs or AmpC enzymes not specifically targeted by the genotypic test in this study, as well as other resistance mechanisms such as porin loss or increased efflux pump activities. It is important to note that CTX belongs to the third-generation cephalosporin (3GC) class commonly used for the treatment of human and animal infections [36]. Third-generation cephalosporin resistance in *Enterobacteriaceae*, including *E. coli*, has been classified as a critical priority on the WHO global priority pathogens list due to the public health threat it poses, its potential for global dissemination and limited treatment options [37]. Several AMR mechanisms can contribute to the development of 3GC resistance in *Enterobacteriaceae*, including *E. coli.* These mechanisms encompass the presence of active efflux pumps and point mutations that lead to a decrease in membrane permeability [9]. However, the most prevalent mechanism observed is the production of ESBLs with the CTX-M type being the most common. These ESBLs are encoded by the *bla*_CTX-M_ gene. The resistance exhibited by *E. coli* isolates carrying *bla*_CTX-M_ and blaTEM genes has significant implications for public health and nutrition in communities where raw milk consumption is prevalent [25]. The presence of these genes in *E. coli* strains found in raw milk suggests the potential for the dissemination of resistance to other pathogenic and commensal bacteria [13]. Moreover, there is a potential for the horizontal transmission of MDR *E. coli* between animal species, in this case cattle and humans, through direct contact or consumption of raw milk [22,38]. Additionally, factors such as cattle movement for trade or to the slaughterhouse play a significant role in the dissemination of AMR *E. coli.* Livestock trade networks, monitoring animal movements and assessment of biosecurity measures during transportation are crucial for understanding the transboundary spread of resistant bacteria [18].

The limitations in this study included firstly, the limited number of milk collection centers (MCC) from which milk samples were collected. Specifically, we were only able to collect samples from four out of the six milk collection centers in the district, as two centers were non-operational during the period of sample collection. This restricted sample size may have impacted the representativeness of our findings and could introduce bias. Secondly, the investigation of additional ESBL-encoding genes, apart from the ones examined in this study, was not possible due to the unavailability of the necessary primers. Furthermore, the lack of resources prevented us from performing whole genome sequencing (WGS). WGS analysis would have provided a broader spectrum of genetic information enabling a more comprehensive characterization of AMR *E. coli* in raw milk.

We recommend that future studies should consider the utilization of WGS to identify other resistance mechanisms. Furthermore, the findings in this study highlight the need for strict monitoring of raw milk production at the primary production level before it is released for consumption.

## 4. Materials and Methods

### 4.1. Study Design and Site

This study was cross-sectional and was conducted from March 2020 to August 2021. The study was conducted in Namwala district in the Southern province of Zambia, located at global positioning system (GPS) coordinates 15°45′ S and 26°27′ E. The district has six milk collection centers (MCC), of which the four functional at the time of sample collection were incorporated in this study (Figure 2). The coding of the four milk collection centers (MCC) was with identification numbers MCC 1, 2, 3 and 4. During this period, raw milk samples that were presented at milk collection centers from different farms were collected before bulking in phases. These sites were selected due to the fact that they are the areas with the highest cattle populations in the country [39]. 

### 4.2. Sample Size and Sampling

The sample size was estimated based on the farmer/farm records at the milk collection center. The submission of milk to the MCC was based on the farmer/farm belonging to the cooperative of that particular MCC. During the sampling period of this study, the number of farmers who submitted milk to the MCC ranged from 200, 80, 100, and 80 for MCC 1, 2, 3, and 4, respectively. Purposive sampling was used to target all of the farmers who submitted a pooled farm raw milk sample to the milk collection center on that particular day. A total of 418 pooled raw milk samples were collected aseptically before bulking from the milking churns in 15 mL falcon tubes. The milk samples were then assigned a sample identification (sample i.d) number based on the MCC they were collected from. The raw milk samples were stored at 4 °C and immediately transported to the laboratory for further processing.

### 4.3. Isolation and Identification of E. coli Species

One milliliter of raw milk sample was inoculated into nine milliliters of sterile Buffered Peptone water (Oxoid, Basingstoke, UK) and incubated at 37 °C for 24 h for enrichment. A sterile loop of 10 µL was used to inoculate the enriched sample onto the surface of MacConkey Agar plates (Oxoid, Basingstoke, UK). The plates were incubated at 37 °C for 24 h, and a few colonies that were lactose fermenting (pinkish, smooth, moist, and circular) were then sub-cultured on Eosin Methylene Blue agar (EMB agar, Oxoid, Basingstoke, UK) for selective identification. The selected colonies that were lactose fermenting on MacConkey agar and had a green metallic sheen on EMB agar were presumed to be *Escherichia coli*. One presumptive colony of *E. coli* was subjected to biochemical analysis (IMViC) using the Sulphur Indole Motility, Urea, Triple Sugar Iron, and citrate test, as well as Gram’s staining. Isolates positive for Indole and motility, had an Acid butt with gas production and acid Slant on TSI but negative for Urea and citrate utilization were confirmed as *E. coli.* For quality control, the ATCC reference strain *E. coli* 25922 was used as a positive control. The isolates were suspended in a 10% glycerol-peptone solution for storage purposes and stored at −80 °C.

### 4.4. Antimicrobial Susceptibility Testing of E. coli Isolates

The antimicrobial susceptibility testing was carried out using the Kirby-Bauer disk diffusion method on Mueller-Hinton agar plates (Oxoid, Basingstoke, UK) according to the Clinical and Laboratory Standards Institute guidelines [40]. Ten antimicrobial drugs were used, as listed in Table 3. The criterion for the selection of antimicrobials was based on the list of antimicrobials imported into Zambia for use as veterinary medicinal products [31]. 

### 4.5. Minimum Inhibitory Concentration (MIC) of Cefotaxime (CTX)

To screen for CTX resistance, broth microdilution was used to quantify CTX resistance [41]. Briefly, *E. coli* isolates resistant to CTX on the disk diffusion method were inoculated on Luria Bertani (LB) agar supplemented with 1 µg/mL CTX and incubated for 18 h at 37 °C. A single colony was then transferred to 3 mL of LB broth supplemented with 1 µg/mL CTX and incubated for 18 h at 37 °C with shaking at 170 rpm. The cultures obtained after 18 h of incubation were then diluted 10^4^-fold and added in triplicates of a serial dilution of cefotaxime in a 96-well plate. The 96-well plates were then incubated for 18 h at 37 °C after which the results were read based on growth in the wells.

### 4.6. DNA Extraction and PCR Identification of Beta-Lactamase-Encoding Genes

Genomic DNA (gDNA) was extracted from pure colonies of the *E. coli* isolates grown on LB media infused with 1 µg/mL CTX using a commercial DNA extraction kit (ZYMO research Quick-DNA TM Fecal/soil microbe miniprep kit) as per manufacturer’s instructions. PCR was performed on 2 µL extracted gDNA in a total reaction volume of 50 μL consisting of 10 ExTaq PCR buffer, 0.25 μL of ExTaq polymerase, 4 μL each of dNTP, 5 μL of forward and reverse primers (CTX-MA, TEM1 and OXA1) targeted at the *bla*_CTX-M_, *bla*_TEM_ and *bla*_OXA_ gene, respectively (Appendix A–Table A1). The PCR conditions were initial denaturation at 94 °C for 7 min followed by 25 cycles of denaturation at 94 °C for 30 s, annealing at 56 °C for 30 s, and extension at 72 °C for 30 s, with a final extension at 72 °C for 5 min. The amplified PCR products were electrophoresed in 1.5% agarose gel before visualization.

### 4.7. Data Analysis

The data obtained from the Kirby-Bauer disk diffusion method were entered onto a Microsoft Excel sheet and exported into WHOnet 2021 software, where the resistant profiles for all of the antibiotics were reported, and frequency tables and graphs were generated. The R statistical program was used to perform further analyses such as frequency distributions and proportions on the AMR data obtained from WHONET software version 5.6.

## 5. Conclusions

This study augments earlier findings that raw milk is a significant source of *E. coli*. Additionally, the findings of multidrug-resistant *E. coli* in raw cow’s milk indicate serious public health concerns. Further, these *E. coli* isolates harbour *bla*_CTX-M_ and *bla*_TEM_ genes that may be responsible for the spread of resistance to other pathogenic and commensal bacteria. These findings highlight the need for strict monitoring of raw milk production at the primary production level before it is released for consumption.

## Figures and Tables

**Figure 1 antibiotics-12-01421-f001:**
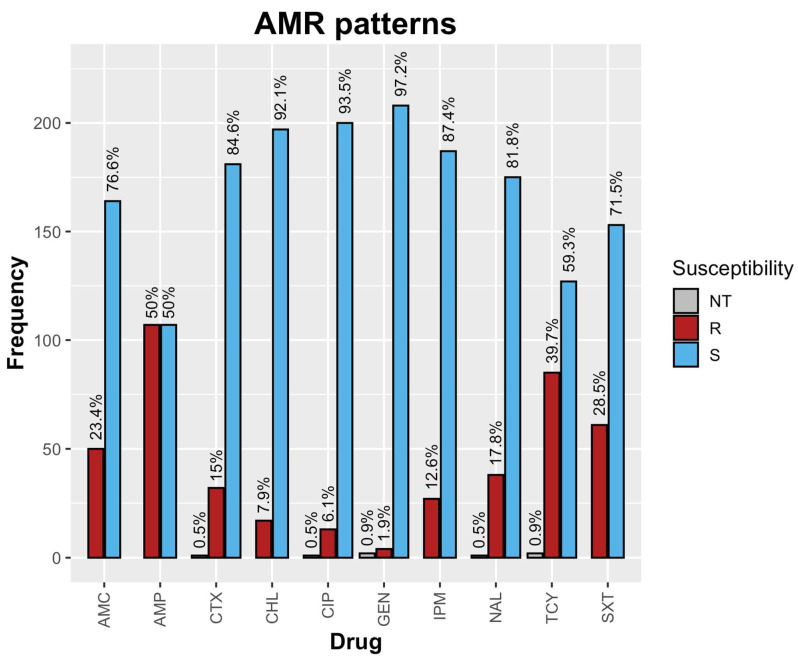
AMR patterns of raw milk *E. coli* isolates. Note: NT = Not tested, R = Resistant, S = Susceptible.

**Figure 2 antibiotics-12-01421-f002:**
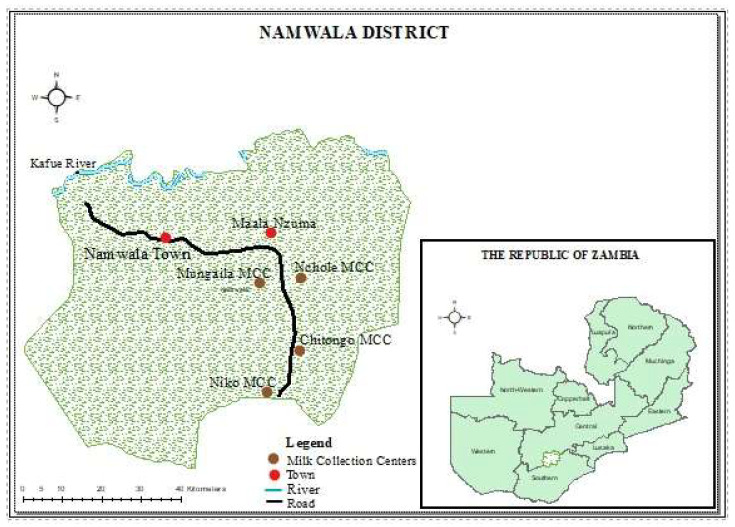
Map of Namwala district, Zambia showing study sites.

**Table 1 antibiotics-12-01421-t001:** Number of cow milk samples positive for *E. coli* (%).

Contamination	MCC (%) *n* = 418				
	MCC 1	MCC 2	MCC 3	MCC 4	95% C.I.
**Non–*E. coli***	100 (54)	48 (64)	38 (39)	18 (30)	44.02–53.61
** *E. coli* **	86 (46)	27 (36)	59 (61)	42 (70)	46.39–55.98

**Table 2 antibiotics-12-01421-t002:** *bla*_CTX-M_ and *bla*_TEM_ presence in CTX-resistant isolates.

Sample ID	Source	CTX MIC	*bla* _CTX-M_	*bla* _TEM_	*bla* _OXA_
158	MCC 1	4	+	−	−
09	MCC 3	8	+	−	−
04	MCC 3	64	−	+	−
14	MCC 1	2	+	−	−
133	MCC 1	32	+	−	−
32	MCC 1	16	−	+	−
41	MCC 3	64	+	+	−
01	MCC 1	4	−	−	−
144	MCC 1	2	−	−	−
11	MCC 4	64	−	−	−
62	MCC 2	4	−	+	−
16	MCC 4	4	−	+	−
122	MCC 1	4	−	+	−
22	MCC 3	16	−	−	−

Note: + = Positive, − = Negative.

**Table 3 antibiotics-12-01421-t003:** Antibiotics and concentrations used.

Antibiotic	Concentration (µg)	Zone Diameter Breakpoints (≤S–≥R)
Amoxicillin-Clavulanic Acid AMC	20	14–17
Ampicillin AMP	10	14–16
Cefotaxime CTX	30	23–25
Chloramphenicol CHL	30	13–17
Ciprofloxacin CIP	5	22–25
Gentamicin GEN	10	13–14
Imipenem IMP	10	20–22
Nalidixic Acid NAL	30	14–18
Tetracycline TCY	30	12–14
Trimethoprim-Sulfamethoxazole SXT	25	11–15

## Data Availability

The data sets analysed during the current study are available from the corresponding author upon reasonable request.

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
