# Peer review of "Multidrug-Resistant Escherichia coli from Raw Cow Milk in Namwala District, Zambia: Public Health Implications"

_antibiotics, 2023, doi:10.3390/antibiotics12091421_

Round 1
Reviewer 1 Report
Thirty two E. coli strains were resistant to cefotaxime by disk diffusion; however, only 14 strains presented low sensitivity to the antimicrobial when the microdilution method was used. Furthermore, only 5 and 6 E. coli strains presented the bla-CTX-M and bla-TEM genes, respectively. How do you explain: first, the difference in the phenotypic results between the two methods used and second, the lack of detection of beta lactamases genes in all these resistant isolates to cefotaxime?
Authors have to meticulously check references, correctly spell the names of microorganisms and genes
I don´t have comments
Reviewer 2 Report
Nevertheless, much work has been done on reporting resistance in E. coli, however, the present article may be important for the Zambia point of view. It reports the rise in MDR E coli recovered from cow milk. This reviewer has following comments to make.
"Data" term should be taken as plural. "Were" should be used for data, instead of was.
Methods should include a schematic map of Zambia showing the concerned district and pinpointing the locations of milk collection units.
The manuscript should cite a few relevant citations from 2023 as well.
Discussion section should discuss the factors which promote resistance in isolates in the nearby countries with reference to Zambia. If Zambia, Tanzania, Namibia, Zimbabwe share some animal movement between their borders?
It is suggested to include the limitations of the study in the Discussion section.
Minor improvements required
